# Vitamin B Mitigates Thoracic Aortic Dilation in Marfan Syndrome Mice by Restoring the Canonical TGF-β Pathway

**DOI:** 10.3390/ijms222111737

**Published:** 2021-10-29

**Authors:** Tzu-Heng Huang, Hsiao-Huang Chang, Yu-Ru Guo, Wei-Chiao Chang, Yi-Fan Chen

**Affiliations:** 1Master Program in Clinical Pharmacogenomics and Pharmacoproteomics, School of Pharmacy, Taipei Medical University, Taipei 11031, Taiwan; harry850325@gmail.com; 2Division of Cardiovascular Surgery, Department of Surgery, Taipei Veterans General Hospital, Taipei 11217, Taiwan; shchang@vghtpe.gov.tw; 3Department of Surgery, School of Medicine, College of Medicine, Taipei Medical University, Taipei 11031, Taiwan; 4Graduate Institute of Cancer Biology and Drug Discovery, College of Medical Science and Technology, Taipei Medical University, Taipei 11031, Taiwan; dinyty@hotmail.com; 5Department of Clinical Pharmacy, School of Pharmacy, Taipei Medical University, No. 250, Wuxing St., Xinyi Dist., Taipei 11031, Taiwan; 6Department of Pharmacy, Wan Fang Hospital, Taipei Medical University, Taipei 116081, Taiwan; 7Integrative Research Center for Critical Care, Wan Fang Hospital, Taipei Medical University, Taipei 116081, Taiwan; 8The Ph.D. Program for Translational Medicine, College of Medical Science and Technology, Taipei Medical University, Taipei 11529, Taiwan; 9Graduate Institute of Translational Medicine, College of Medical Science and Technology, Taipei Medical University, Taipei 11031, Taiwan; 10International Ph.D. Program for Translational Science, College of Medical Science and Technology, Taipei Medical University, Taipei 11031, Taiwan

**Keywords:** Marfan syndrome, fibrillin 1 mutation, thoracic aortic aneurysm (TAA), thoracic aortic dissection (TAD), vitamin B, folate-methionine cycle

## Abstract

Thoracic aortic aneurysm (TAA) formation is a multifactorial process that results in diverse clinical manifestations and drug responses. Identifying the critical factors and their functions in Marfan syndrome (MFS) pathogenesis is important for exploring personalized medicine for MFS. Methylenetetrahydrofolate reductase *(MTHFR)*, methionine synthase (*MTR)*, and methionine synthase reductase (*MTRR)* polymorphisms have been correlated with TAA severity in MFS patients. However, the detailed relationship between the folate-methionine cycle and MFS pathogenesis remains unclear. *Fbn1*^C1039G/+^ mice were reported to be a disease model of MFS. To study the role of the folate-methionine cycle in MFS, *Fbn1*^C1039G/+^ mice were treated orally with methionine or vitamin B mixture (VITB), including vitamins B6, B9, and B12, for 20 weeks. VITB reduced the heart rate and circumference of the ascending aorta in *Fbn1*^C1039G/+^ mice. Our data showed that the *Mtr* and *Smad4* genes were suppressed in *Fbn1*^C1039G/+^ mice, while VITB treatment restored the expression of these genes to normal levels. Additionally, VITB restored canonical transforming-growth factor β (TGF-β) signaling and promoted *Loxl1*-mediated collagen maturation in aortic media. This study provides a potential method to attenuate the pathogenesis of MFS that may have a synergistic effect with drug treatments for MFS patients.

## 1. Introduction

Marfan syndrome (MFS) is an inherited connective tissue disorder that is often caused by the mutation of fibrillin 1 (Fbn1) and the consequent extracellular matrix (ECM) degeneration [1,2]. Because thoracic aortic aneurysm (TAA) and thoracic aortic dissection (TAD) are the leading causes of mortality in MFS patients, aortic sizes are closely followed during diagnosis and treatment [3]. TAA, however, is a multifactorial condition that has a wide variety of clinical severities. Researchers have studied different aspects of MFS pathology for decades to explore personalized therapies [4,5].

It has been assumed for a long time that Transforming growth factor β (TGF-β) signaling plays a central role in MFS pathology. J. Habashi first reported that TGF-β signaling was upregulated in Marfan patients and 12-month-old Marfan mice (Fbn1C1039G/+). Furthermore, TAA in the Fbn1C1039G/+ mouse model was rescued by either directly inhibiting TGF-β or suppressing the downstream noncanonical TGF-β signaling pathway [6,7]. Angiotensin II receptor blockers (ARBs) and statins attenuated TAA in preclinical studies by suppressing the noncanonical TGF-β signaling pathway, and these are promising therapies for MFS patients [7,8]. However, recent studies examined the origin of TAA pathology. H. Wei et al. examined the causative relationship between the TGF-β signaling pathway and TAA in 8-week-old Fbn1C1039G/+ mice and found that both canonical and noncanonical TGF-β signaling pathways showed no differences compared to wild-type mice [9]. On the other hand, Jason R. Cook and colleagues reported that a TGF-β neutralizing antibody was harmful at the early stage of Marfan syndrome and proposed that TGF-β might play completely different roles during disease progression [10]. The canonical TGF-β signaling pathway, on the other hand, plays an entirely different role in TAA pathology. The Smad4-haploinsufficient MFS mouse model (Smad4+/−: Fbn1C1039G/+ genotype) showed a larger aortic aneurysm and higher mortality rate than the MFS mouse model [11]. Smad4 deficiency in smooth muscle cells (SMCs) induced TAA via the upregulation of proteases, cathepsin S, and metalloproteinase 12 (Mmp12) [12]. Therefore, the upregulation of the Smad4-dependent TGF-β signaling pathway in SMCs protected against aortic aneurysm formation and dissection. Furthermore, recent studies showed that the expression of lysyl oxidases, downstream of both canonical and noncanonical TGF-β pathways, was highly correlated with the pathogenesis of TAA. Lox mutation causes severe TAA and TAD in clinical patients [13,14]. In an MFS mouse model, the inhibition of Lox by β-aminopropionitrile (BAPN) rapidly caused aortic disintegration and dissection; therefore, lysyl oxidases reserved aortic integrity by promoting ECM maturation and cross-linking [15].

The folate cycle has been correlated with cardiovascular diseases because it can regulate reactive oxygen species (ROS) production and modify DNA methylation [16,17]. One of the folate-methionine cycle metabolites, homocysteine, acted as a partial agonist of angiotensin II receptor in abdominal aortic aneurysms [18]. In regard to the folate cycle in MFS TAA, the mechanism of action remains unclear. Clinical studies have shown that polymorphisms in folate cycle enzymes, methylenetetrahydrofolate reductase (MTHFR), methionine synthase (MTR), and methionine synthase reductase (MTRR), are related to aneurysm severity and dissection potential in MFS patients [19]. Although they hypothesized that hyperhomocysteinemia is the cause of clinical diversity, this hypothesis failed to be validated in a hyperhomocysteinemia-MFS mouse model induced by a cobalamin-restricted diet [20,21].

This is the first study investigating the effect of methionine-induced hyperhomocysteinemia on TAA in MFS mice. Additionally, we firstly examined the interaction between the folate cycle and MFS pathology by treating MFS mice with a novel vitamin B mixture (VITB). By using a precise aortic tissue slicing technique and quantitative PCR, we could determine the therapeutic effect of VITB and the potential mechanism of this action. The result suggested that a potential supplement, vitamin B complex, could synergize the effects of clinic-use drugs on MFS patients.

## 2. Results

### 2.1. TAA Was Associated with Insufficient Smad4, Mtr, and Mtrr in Fbn1C1039G/+ (MFS) Mice

To assess the disease phenotype, we measured the size of both the aortic root and ascending aorta. We also measured the aortic roots of 6-month-old MFS and WT mice by echocardiography. The sinus diameter of MFS mice was significantly longer than that of WT mice (Appendix A). To measure the proximal ascending aortic size, thoracic aortas from the aortic root to the proximal ascending aorta were embedded in paraffin, the direction was adjusted and the location was confirmed. The ascending aortic diameter of MFS mice was significantly larger than that of WT mice (Appendix A). Next, we investigated the association between TAA pathological genes and folate cycle genes with qPCR. The TGF-β signaling pathway is the central pathological pathway in MFS. A previous report indicated that polymorphisms in folate cycle enzymes, MTHFR, MTR, and MTRR, were related to aneurysm severity and dissection potential in MFS patients [19]. The results showed that Smad4, Mtr, and Mtrr were suppressed in MFS TAAs compared to WT TAAs. The suppression of Smad4 in MFS TAAs was further confirmed by Smad4 immunostaining and Smad4-regulated gene expression, such as Serpine1 (Figure 1A,B). The RNA expression levels of Mtr and Mtrr were both positively correlated with Smad4 expression levels in MFS mouse TAA tissue (Figure 1C).

### 2.2. The Effects of Methionine and VITB Treatment on Plasma Homocysteine and Heart Rate in Fbn1C1039G/+ (MFS) Mice

It has been reported that Mtr and Mtrr gene polymorphisms are associated with the severity of TAA in MFS patients, and high levels of plasma homocysteine are a key factor [19]. However, we observed no changes in plasma homocysteine in MFS mice when we observed alterations in Mtr and Mtrr expression levels. To study the effects of homocysteine on TAA severity, we induced hyperhomocysteinemia in both MFS and WT mice with treatment with 1% methionine in drinking water (Figure 2A). To address the folate metabolic cycle, we further designed VITB, consisting of vitamins B6, B9, and B12, and treated MFS mice with VITB to study the potential effect on TAA formation. At the same time, ATE was chosen for comparison because it can reduce cardiac afterload and be used in clinical practice. We found that methionine treatment increased the heart rate of MFS mice but not WT mice. Both VITB and ATE treatment reduced heart rate, and they could be combined in MFS mice (Figure 2B). Moreover, the blood pressure was also recorded using the tail-cuff method (Appendix A). Although the blood pressure in the methionine treatment group was slightly higher at systolic blood pressure compared to the other groups, there was no significant difference (Appendix A).

### 2.3. The Effects of Methionine and VITB Treatment on TAA in Fbn1C1039G/+ (MFS) Mice

We examined the length of the external elastic lamina (EEL) in the ascending aortas of mice (Figure 3A). VITB treatment significantly reduced the circumference of the Marfan aortic aneurysm, while neither methionine nor ATE treatment made differences (Figure 3B). The aortic wall thickness was mathematically calculated by the lengths of the EEL and internal elastic lamina. The aortic elasticity was assessed by quantifying elastic fiber breakdown with VVG staining. MFS mice had significantly thicker aortic media and less elasticity than WT mice. However, none of the regimens restored the media thickness and elasticity (Figure 3C,D).

### 2.4. VITB Restored Smad4 Expression and Promoted Loxl1-Mediated Collagen Deposition in TAA

Along with elastic fibers, collagen fibers provide structural strength in the thoracic aorta. Some authors have reported that lysyl oxidases stabilize TAA by cross-linking ECM proteins and mediating collagen deposition in MFS patients and mice. Therefore, we stained sections with picrosirius stain and photographed the slides under polarized light to stain media collagen fibers. Our results show that MFS mice have significantly more collagen deposition areas in the aortic media than WT mice. VITB treatment elevated the collagen deposition proportion compared to the vehicle control (Figure 4A,B). Lysyl oxidase is one of the downstream effectors of the TGF-β signaling pathway. We examined the gene expression of the TGF-β signaling pathway and folate cycle genes that showed strong associations in our previous data. VITB treatment restored Smad4 expression and elevated Loxl1 expression compared to the vehicle control. The expression level of Loxl1 was positively correlated with the proportion of collagen deposition in the aortic media (Figure 5A,B).

## 3. Discussion

The TGF-β signaling pathway plays a critical role in TAA formation, but the origin of MFS pathology remains controversial [22]. The effects of TGF-β regulation also seem to be phase-dependent changes [10]. Many factors involved in the disease progression of TAA, including folate cycle enzymes, are still unclear [19]. In this study, we first observed a decrease in Smad4 expression in 6-month-old MFS mice. We found a positive correlation between Smad4 gene expression and genes involved in the folate cycle. These findings provide us with new information to explain patient clinical diversity along with gene polymorphisms in folate cycle enzymes. Our data showed that VITB treatment elevated the expression of smad4 and genes in the folate cycle. However, the detailed mechanism requires further investigation.

We believe that elevated homocysteine contributes little to the pathogenesis of TAAs, even though it is a major risk factor for inducing abdominal aortic aneurysms in other disease models [20]. Both methionine- and a cobalamin-restricted diet-induced hyperhomocysteinemia in Fbn1C1039G/+ mice but did not accelerate aortic dilation. Additionally, our dosage of VITB treatment was not high enough to lower the plasma homocysteine concentration but still mitigated the aortic dilation size. These results suggest that the therapeutic effects of VITB were homocysteine independent.

Considering the smaller size of TAAs and greater collagen deposition in the aortic media after VITB treatment, we hypothesized that the canonical TGF-β pathway played a protective role. In a Smad4-haploinsufficient MFS mouse model (S4+/−: Fbn1C1039G/+), mice develop TAD along with noncanonical TGF-β pathway activation [11]. In a smooth muscle cell-specific Smad4-deficient mouse model, a lack of Smad4 protein induced the overexpression of MMP12 and decreased the expression of ECM proteins and lysyl oxidase [12]. In our study, we observed the upregulation of MMP12 in MFS mice, which was downregulated by VITB treatment (data not shown). We did not find any difference in elastin breakdown between the VITB treatment and vehicle control groups. A possible explanation is that elastic fibers have a long turnover time. Thus, the repair of elastic fibers takes a longer time than collagen fibers once TAAs have formed. Although changes in elasticity were not observed, the elevation of the Smad4 and Loxl1 genes implied that the aortic structure was strengthened. Previous reports indicated that mutations in lysyl oxidase are associated with aortic dissection [13,14]. Our results showed that lysyl oxidase was positively correlated with the collagen proportion in the aortic media. Although some studies proposed that collagen deposition makes the aortic media stiffer and worsens the biomechanical properties, a lack of mature collagen fibers induced earlier onset of aortic dissection in diseases such as Ehlers–Danlos syndrome [23,24]. Furthermore, the inhibition of collagen deposition by inhibiting lysyl oxidase exacerbated MFS aortic aneurysm and caused aortic dissection [15]. Therefore, we believe that collagen stabilizes aortic aneurysms in pathological situations. Thus, VITB treatment has potential therapeutic effects on stabilizing TAA and might be able to prevent dissection by restoring the canonical TGF-β pathway (Figure 6).

## 4. Conclusions

This study described the positive correlation between folate-methionine genes and canonical Tgfβ genes in ascending aorta of Fbn1C1039G/+ mice. And Vitamin B including B6, B9, B12 affected TAA pathology by regulating collagen maturation but not global homocysteine concentration. The specific mechanism of vitamin B interaction with the extracellular matrix still needs further investigation. However, this study provides a potential method to alleviate the thoracic aortic dilation in MFS.

## 5. Materials and Methods

### 5.1. Experimental Animals

Fbn1C1039G/+ mice were purchased from Jackson Laboratory (B6.129-Fbn1tm1Hcd/J, #012885). Experiments were performed with at least seven male Fbn1C1039G/+ mice and their littermate controls (wild-type, WT) in each group. The animal protocol was approved by the Institutional Animal Care and Use Committee of Taipei Medical University and National Defense Medical Center.

### 5.2. Treatment Groups

To test whether hyperhomocysteinemia affects TAA manifestation, we used 1% methionine drinking water to induce the model [25]. Moreover, Atenolol was added to the drinking water (0.5 g/L) as a standard drug treatment [26]. For vitamin B dosage, a meta-analysis concluded that a folate-based therapy effectively reduces plasma homocysteine levels in humans [27]. Using the dosage conversion method, we determined the following drug dosage and experiment groups [28].

Four-week-old WT mice (*n* = 8–11 per group) were treated orally with either: (1) methionine (≥98% purity; 1% in drinking water) (M9625, Sigma-Aldrich, St Louis, MO, USA) or (2) vehicle control for 20 weeks. Four-week-old Fbn1C1039G/+ mice (*n* = 7–15 per group) were treated orally with either (1) methionine (1% in drinking water), (2) methionine (1% in drinking water) and VITB (≥97% purity, 25.7 μg/day folate via gavage; ≥96% purity, 0.865 mg/L cyanocobalamin in drinking water; and ≥98% purity, 86.5 mg/L pyridoxine in drinking water) (F7876, C3607, P5669, Sigma-Aldrich), (3) methionine (1% in drinking water) and atenolol (ATE; ≥98% purity, 0.5 g/L in drinking water) (A7655, Sigma-Aldrich), (4) methionine (1% in drinking water), ATE (0.5 g/L in drinking water), and VITB (25.7 μg/day folate via gavage; 0.865 mg/L cyanocobalamin in drinking water; and 86.5 mg/L pyridoxine in drinking water) (F7876, C3607, P5669, Sigma-Aldrich), or (5) vehicle control for 20 weeks. Tissues were harvested from mice at the age of 24 weeks.

### 5.3. Echocardiography

Mouse thoracic aortas were examined by echocardiography at the age of 24 weeks. Echocardiography was performed by Taiwan Mouse Clinic. Mice were anesthetized by the inhalation of isoflurane, and chest hair was removed. The parasternal long-axis view was examined by ultrasound (Vevo3100). The aortic root diameter was measured at diastolic intervals.

### 5.4. Tail-Cuff Pulse Rate Experiment

Pulse rate experiments were performed by the Taiwan Mouse Clinic when mice were 24 weeks old. Mice were transferred into a quiet and independent room for the experiment. Using a BP-2000 Series II Blood Pressure Analysis System, pulse rate and blood pressure were measured from mouse tails without anesthesia or any invasive methods.

### 5.5. Tissue Sampling

Mouse blood (500–700 μL) was collected in a tube with 100 μL EDTA (0.05 M). After centrifugation at 3000 rpm for 10 min at 4 °C, plasma was collected and aliquoted before storage at −80 °C. To collect the aorta tissue, 5 mL ice-cold saline solution was used for perfusion from the left ventricle to the renal artery. The heart, thoracic aorta, and abdominal aorta were carefully excised with a scalpel, and the fresh tissue was placed in a dish filled with ice-cold saline solution. Tissues from the distal ascending aorta to the aortic arch were excised for RNA analysis. All branches and adventitia were removed from aortic tissues, and then the aortic tissues were placed in RNAlater solution (AM7030, Thermo Fisher, Waltham, MA, USA) and stored at −20 °C. For histological analysis, tissues from the aortic root to the proximal aorta were carefully separated from the heart tissue. The aorta was kept as intact as possible and placed in 10% formalin (Burnett) for 24 h of fixation. Well-fixed aortae were stored in 70% ethanol for at least 24 h before dehydration and paraffin embedding.

### 5.6. RNA Analysis

RNA was extracted from the ascending and aortic arch tissues using TRIzol™ Reagent (15596018, Invitrogen, Carlsbad, CA, USA) according to the manufacturer’s instructions. Complementary DNA (cDNA) was synthesized using a High-Capacity cDNA Reverse Transcription Kit (4368813, Applied Biosystems, Waltham, MA, USA). Real-time (RT)-quantitative PCR (qPCR) was performed with Power SYBR™ Green PCR Master Mix (4367359, Applied Biosystems™) and analyzed with a QuantStudio3 Real-Time PCR system (Applied Biosystems) under standard conditions. All amplification reactions were carried out in triplicate for each RNA sample. The amount of total input cDNA was calculated using hypoxanthine-guanine phosphoribosyl transferase (Hprt) as an internal control.

### 5.7. Paraffin Cross-Section and Verhoeff–Van Gieson Staining

The orientation of the aortic root significantly affects the outcomes of histological analysis. We embedded aortic tissue in a 4% agarose gel before paraffin embedding. Three-micrometer paraffin cross-sections were examined slide-by-slide during sectioning to correct the vertical orientation of aortic tissues. We precisely determined the position of the sinotubular junction (STJ) and confirmed the correct size of each section [29]. Each slide contained three cross-sections with 9 μm intervals at the STJ section for further analysis. Modified Verhoeff-van Gieson (VVG) staining reagents (26369-01~05, Electron Microscopy Science, Hatfield, PA, USA) were used to stain the elastic fibers in aortic tissues. The lengths of the internal and external elastic lamina (EEL) were measured by ImageJ software. The aortic diameter and aortic media thickness were calculated with formulas. The aortic elasticity was examined by calculating the number of elastic fibers that had broken down. The presence of two close elastin free ends was considered one break.

### 5.8. Picrosirius Red Staining and Collagen Deposition Quantification

Three-micrometer paraffin cross-sections were subjected to picrosirius red staining (24901, Polysciences Inc., Warrington, PA, USA) by standard procedures. Successfully stained slides were photographed under a microscope with a polarizing filter. The area of birefringence in the aortic media was quantified by ImageJ software.

### 5.9. Smad4 Immunostaining

Antigen retrieval buffer (pH = 6, DakoCytomation, Glostrup, Denmark) was used to unmask the antigen. Immunohistochemistry (IHC) staining was performed by incubating sections with Smad4 primary antibodies (1:100) for 18–24 h at 4°C and detected using biotinylated secondary antibodies and an LSAB Kit (DakoCytomation).

### 5.10. Homocysteine Colorimetric Assay

Plasma samples were obtained from 24-week-old mice. Samples with hemolysis or precipitation were excluded. Plasma homocysteine levels were determined by a colorimetric assay kit (E-BC-K143, Elabscience, Huston, Tx., USA) according to the manufacturer’s instructions.

### 5.11. Statistical Analysis

The results are presented as the mean ± standard deviation (SD) of at least three independent samples. Comparisons between two groups were carried out using two-tailed, unpaired Welch’s *t*-tests. Statistical differences were considered significant when *p* < 0.05. Asterisks indicate * *p* < 0.05; ** *p* < 0.01; and *** *p* < 0.001.

## Figures and Tables

**Figure 1 ijms-22-11737-f001:**
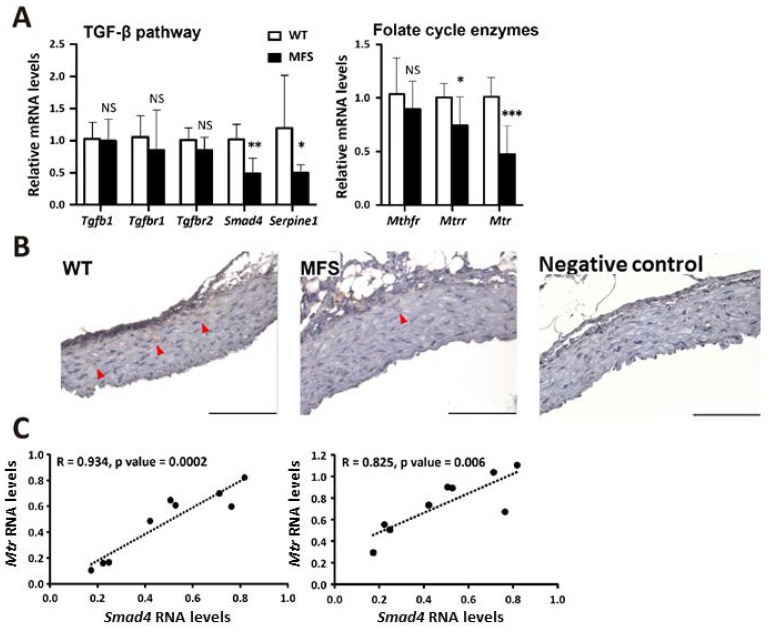
Positive correlation between insufficient Smad4, Mtr, and Mtrr RNA expression in 6-month-old MFS ascending aortas. (**A**) Real-time qPCR analysis of genes involved in the TGF-β pathway and the folate cycle metabolism in aortic tissues with adventitia removed (WT, *n* = 6; MFS, *n* = 9). (**B**) Smad4 immunostaining of proximal ascending aorta. Scale bar, 100 μm. (**C**) Positive correlation between Smad4, Mtr, and Mtrr RNA expression of MFS aortic tissue (*n* = 9). Data are presented as the mean ± SD. NS, no significant difference. * *p* < 0.05, ** *p* < 0.01, *** *p* < 0.001.

**Figure 2 ijms-22-11737-f002:**
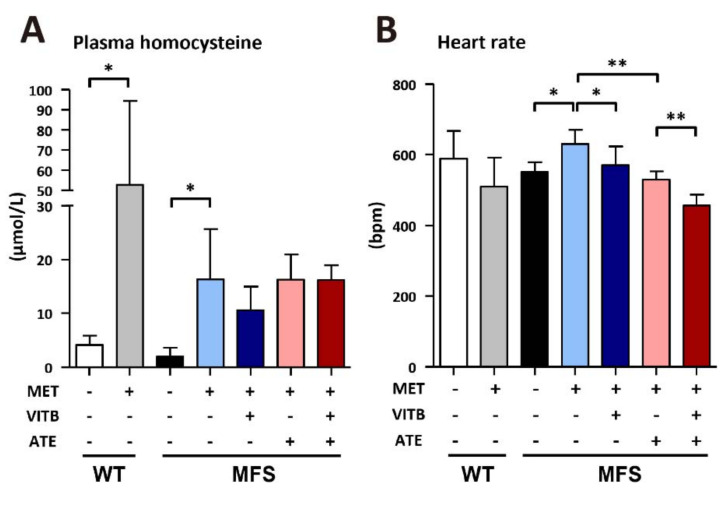
The effect of methionine and VITB treatment on plasma homocysteine and heart rate. (**A**) Plasma homocysteine in WT and MFS mice with different regiments (*n* = 3–5 per group). (**B**) Heart rate counts were measured by tail-cuff in WT and MFS mice with different regiments (*n* = 5–11 per group). Data are presented as the mean ± SD. * *p* < 0.05, ** *p* < 0.01.

**Figure 3 ijms-22-11737-f003:**
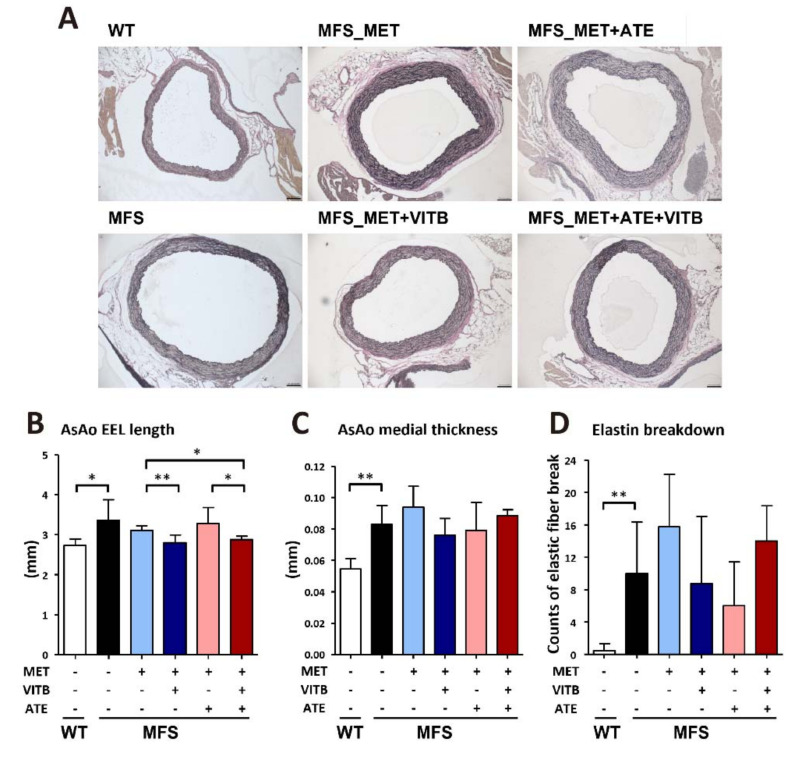
The effects of methionine and VITB treatment on thoracic aortic aneurysm in 6-month-old MFS mice. (**A**) Representative Verhoeff-Van Gieson stain section of MFS and WT ascending aorta (AsAo). Elastin was stained in black; collagen was stained in red. Scale bar, 100 μm. (**B**) AsAo external elastic laminar (EEL) length in WT and MFS mice with different regiments (*n* = 3−9 per group). (**C**) AsAo medial thickness in WT and MFS mice with different regiments (*n* = 3–9 per group). (**D**) The counts of elastic fiber break in WT and MFS mice with different regiments (*n* = 3–6 per group). Data are presented as the mean ± SD. * *p* < 0.05, ** *p* < 0.01.

**Figure 4 ijms-22-11737-f004:**
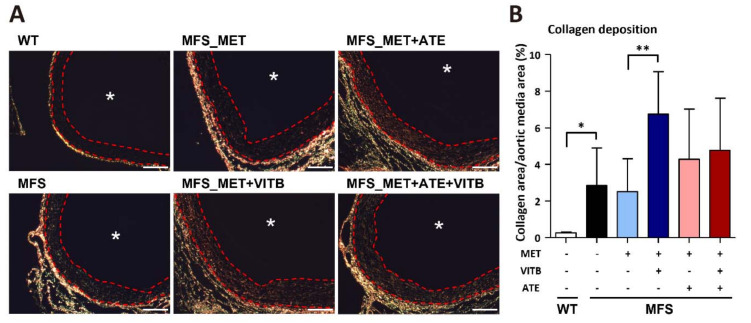
VITB increased collagen deposition in ascending aortic aneurysm of MFS mice. (**A**) Representative Picro-Sirius stain section of ascending aorta under polarized light. Collagen fibers present birefringence under polarized light. Red dashed line indicates aortic media boundary. Asterisk indicates aorta luminal side. Scale bar, 100 μm. (**B**) Quantification of collagen deposition in WT and MFS mice with different regiments (*n* = 3–6 per group). Data are presented as the mean ± SD. * *p* < 0.05, ** *p* < 0.01.

**Figure 5 ijms-22-11737-f005:**
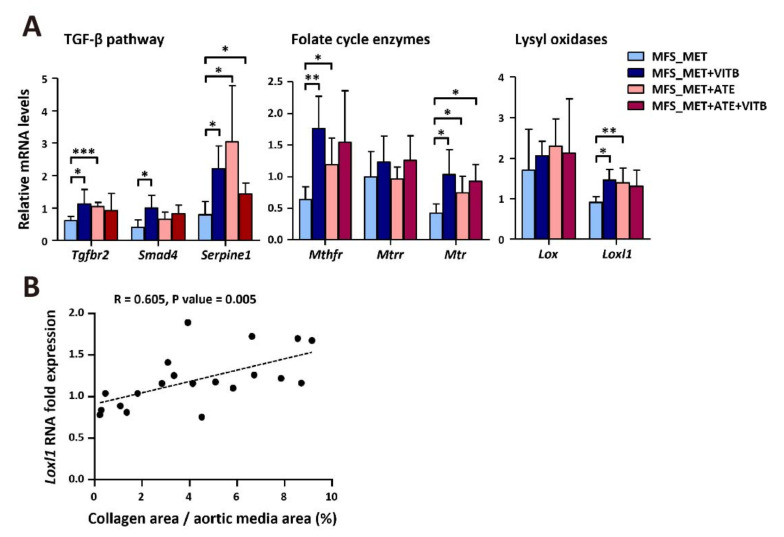
VITB restored Smad4 expression and promoted Loxl1-mediated collagen deposition in TAA. (**A**) Real-time qPCR analysis of gene expression of TGF-β pathway, folate cycle metabolic enzymes, and lysyl oxidases in aortic tissues with adventitia removed (*n* = 4–6 per group). (**B**) Positive correlation between Loxl1 RNA expression and collagen deposition in aortic media of all groups (*n* = 20). Data are presented as the mean ± SD. * *p* < 0.05, ** *p* < 0.01, *** *p* < 0.005.

**Figure 6 ijms-22-11737-f006:**
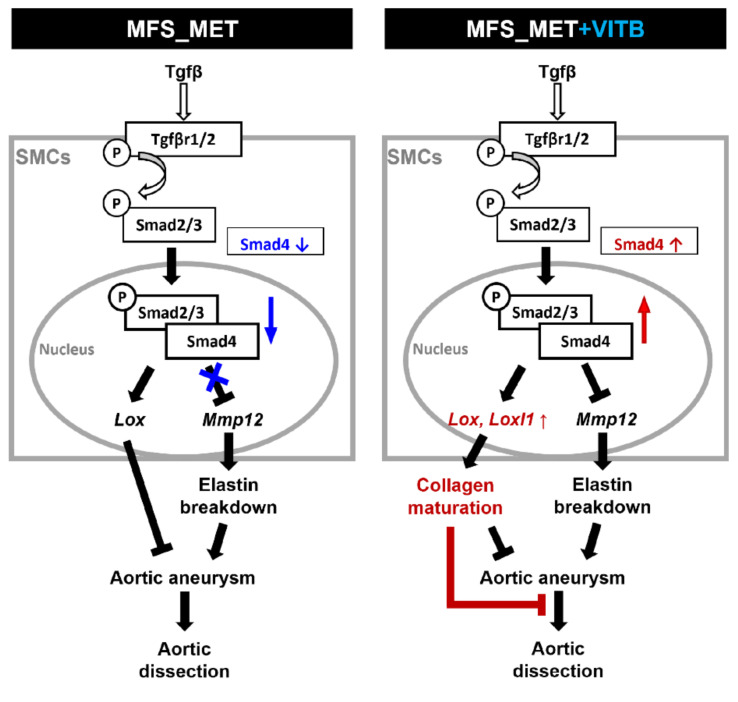
VITB mitigates TAA dilation in MFS mouse model. TGF-β pathway plays a central role in MFS TAA formation. The insufficiency of Smad4 results in lacking control of metalloproteinase and less ECM maturation in 6-month-old MFS mice (left). VITB treatment (right) rescued smad4 expression to the normal level and elevated lysyl oxidase expression. Collagen maturation induced by lysyl oxidases could strengthen the aortic wall and slow down the TAA dilation. Collagen maturation might provide structural integrity and protect the aorta from dissection.

## Data Availability

Not applicable.

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
