# Peer review of "Vitamin B Mitigates Thoracic Aortic Dilation in Marfan Syndrome Mice by Restoring the Canonical TGF-β Pathway"

_ijms, 2021, doi:10.3390/ijms222111737_

Round 1

Reviewer 1 Report

The study by Huang et al. investigated the effects of methionine and Vit B treatment on the development of experimental thoracic aneurysm. The study was performed in  Fbn1C1039G/+ mice with Marfan syndrome which were treated with different compounds for 20 weeks. Echocardiographic, histological and molecular biology showed the changes in  aortic diameter and structure, and modification in TGF beta pathway after VIT B treatment . The authors concluded that modulating the folate-methionine cycle with a VIT B mitigated aortic dilation by restoring the canonical TGF-β pathway and promoting Loxl1- 43 induced collagen maturation.

This is an interesting but preliminary study. The main limitation of the study is that the authors did not use adequate methods for proving their hypothesis.

Comments:

  1. The external aortic diameter was investigated using ultrasound. The measurements should be presented in the table at different time points in comparison with vehicle treated group.
  1. Blood pressure parameters are important. Tail - cuff measurements can usually provide this information.
  2. The expression of the proteins is demonstrated only by using immunohistochemistry. These data should be confirmed by other molecular biology methods (Western Blotting), at least for the key factors.
  3. The dose of the drugs should be explained.
  4. The blood samples were collected (Methods) but the data are not presented. Please correct.
  5. Overall, more quantitative methods should be used. For example, to determine collagen content, the concentrations of hydroxyproline could be measured. Many data provide only mechanistically associations and do not confirm pathways regulations.

Author Response

About the reviewers’ comments, we answered point-by-point as follow and also showed in the revised manuscript by yellow highlight in Microsoft Word.

Comments:

1. The external aortic diameter was investigated using ultrasound. The measurements should be presented in the table at different time points in comparison with vehicle treated group.

Response: Thanks for your suggestion. Because the ultrasound measurement has been constrained by animal room space and Instrument used, we had to choose final point.

2. Blood pressure parameters are important. Tail - cuff measurements can usually provide this information.

Response: Thanks for your comment. We have added the blood pressure on the Supplementary figure 2 according to the previous comment.

3. The expression of the proteins is demonstrated only by using immunohistochemistry. These data should be confirmed by other molecular biology methods (Western Blotting), at least for the key factors.

Response: Thanks for your suggestion. In order to investigate the aorta, we planned to analyze as the attached figure showed. The aorta sub-region was only enough for IHC and RNA analysis, there was insufficient sample to implement Western blotting. IHC analysis was used to observed the protein expression pattern in aorta.

4. The dose of the drugs should be explained.

Response: In text: 4.2 Treatment groups, we have added the information required as explained above.

5. The blood samples were collected (Methods) but the data are not presented. Please correct.

Response: The blood was collecting for detecting homocysteine concentration in Figure 2A.

6. Overall, more quantitative methods should be used. For example, to determine collagen content, the concentrations of hydroxyproline could be measured. Many data provide only mechanistically associations and do not confirm pathways regulations.

Response: Thanks for your suggestion. Because of the limited tissue for analysis, we tried to use RNA expression and the quantification of IHC or specific staining and suggest our conclusion. For example, the Picro-Sirius stain was detected and quantified using polarized light microscopy in Figure 4A. (Reference: https://doi.org/10.1016/j.mex.2015.02.007)

Reviewer 2 Report

General comments

The topic of the submitted manuscript well fits the aim and scope of International Journal of Molecular Sciences, but it has to be reorganised, since the Abstract section has to be summarised, the originality of the present work has to be highlighted, the Experimental section has to be reported before the Results one, and the Conclusions section has to be missed and has to be added.

A deep English grammar and language revision is strongly suggested.

Specific comments and remarks are listed below point by point.

Abstract

The Abstract section is too long. It has to be summarised in order to make it more immediate and concise, highlighting the main findings.

Some results could be reported in the Conclusions section that has to be introduced, since it was missed.

  1. Introduction

    - The incipit “Marfan syndrome (MFS) is an inherited connective tissue disorder that is often 50 caused by the mutation of Fibrillin 1 (Fbn1) and the consequent extracellular matrix (ECM) 51 degeneration” needs appropriate references.

- Similarly, the following sentence “TAA, however, is a multifactorial condition that has a wide variety of clinical severities. Researchers have studied different aspects of MFS pathology for decades to explore personalized therapies.” has to be supported with suitable references.

- The Authors have to better highlight the originality and added value of their work not only with respect to their previous paper, but also with respect to the other previous literature reports about the same topic.

- It is recommendable to indicate at the end of the Introduction section the main employed characterisation techniques in order to achieve the indicated purpose.

  1. Results

2.1 TAA was associated with insufficient Smad4, Mtr, and Mtrr in Fbn1C1039G/+ (MFS) mice

- The consideration “has been reported that Mtr and Mtrr gene polymorphisms are associated with the severity of TAA in MFS patients, and high levels of plasma homocysteine are a key factor.” needs suitable references.

  1. Materials and Methods

- This section has to be reported between the Introduction and the Results ones.

- More details, such as the purity, for all the used chemical and reagents have to be added.

  1. Conclusions

The Conclusions section has to be  missed. Please add it.

Author Response

About the reviewers’ comments, we answered point-by-point as follow and also showed in the revised manuscript by yellow highlight in Microsoft Word.

Comments:

The topic of the submitted manuscript well fits the aim and scope of International Journal of Molecular Sciences, but it has to be reorganised, since the Abstract section has to be summarised, the originality of the present work has to be highlighted, the Experimental section has to be reported before the Results one, and the Conclusions section has to be missed and has to be added.

A deep English grammar and language revision is strongly suggested.

Response: Thanks for your comments. This manuscript was edited by Springer Nature Author Services. Please find the attached file (Certificate).

Specific comments and remarks are listed below point by point.

Abstract

The Abstract section is too long. It has to be summarised in order to make it more immediate and concise, highlighting the main findings.

Response: The abstract section modified throughout the text according to the comment.

Some results could be reported in the Conclusions section that has to be introduced, since it was missed.

1. Introduction

- The incipit “Marfan syndrome (MFS) is an inherited connective tissue disorder that is often 50 caused by the mutation of Fibrillin 1 (Fbn1) and the consequent extracellular matrix (ECM) 51 degeneration” needs appropriate references.

Response: Thanks for your comments. We have added the reference required. [1,2]

- Similarly, the following sentence “TAA, however, is a multifactorial condition that has a wide variety of clinical severities. Researchers have studied different aspects of MFS pathology for decades to explore personalized therapies.” has to be supported with suitable references.

Response: Thanks for your comments. We have added the reference required. [4,5]

- The Authors have to better highlight the originality and added value of their work not only with respect to their previous paper, but also with respect to the other previous literature reports about the same topic.

Response: Thanks for your comments. We added our value in the last paragraph of Discussions.

- It is recommendable to indicate at the end of the Introduction section the main employed characterisation techniques in order to achieve the indicated purpose.

Response: Thanks for your suggestions. We have modified it.

2. Results

2.1 TAA was associated with insufficient Smad4, Mtr, and Mtrr in Fbn1C1039G/+ (MFS) mice

- The consideration “has been reported that Mtr and Mtrr gene polymorphisms are associated with the severity of TAA in MFS patients, and high levels of plasma homocysteine are a key factor.” needs suitable references.

Response: Thanks for your comments. We have added the reference required. [19]

3. Materials and Methods

- This section has to be reported between the Introduction and the Results ones.

Response: The IJMS publish template suggest put Materials and Methods at last section.

- More details, such as the purity, for all the used chemical and reagents have to be added.

Response: Thanks for your suggestion. We have added the information required. (Text: 4.2 Treatment groups)

4. Conclusions

The Conclusions section has to be missed. Please add it.

Response: We have added the Conclusions section.

Round 2

Reviewer 1 Report

Since the authors refer to the results obtained via immunohistology and microscopy, these results should be properly presented:

-The structure of the aorta schould be detectable in all histological figures

- please also provide IHC in larger magnification

- negative control by IHC is missing

The authors provided additional blood pressure parameters which are not modified. How can you explain the changes in the heart rate after Vit B treatment?

Author Response

Since the authors refer to the results obtained via immunohistology and microscopy, these results should be properly presented:

  1. The structure of the aorta schould be detectable in all histological figures

Response: Thanks for your comments. We have modified the Materials and Methods section required. In order to get the exact position of the proximal ascending aortic slide, we check time and again through microscopic examination. (Materials and Methods 5.7)

  1. please also provide IHC in larger magnification.

Response: Thanks for your suggestions. The Smad4 IHC were observed using the maximum magnification 400X.

  1. negative control by IHC is missing

Response: Thanks for your suggestions. We have added the negative control in Figure 1B as required.

  1. The authors provided additional blood pressure parameters which are not modified. How can you explain the changes in the heart rate after Vit B treatment?

Response: Thanks for your comments. Our data showed that atenolol and vitamin B mixture only affect the heart rate but not blood pressure. A possible explanation why atenolol didn’t lower the blood pressure might be due to the dosage we used. Beta1 blockers sometimes show stronger and dose-dependent effect on heart rate but blood pressure [1]. As for vitamin B mixture, we didn’t investigate more details about its mechanism on lowering heart rate.

Ref.

  1. Wong GW, Boyda HN, Wright JM: Blood pressure lowering efficacy of beta-1 selective beta blockers for primary hypertension. Cochrane Database Syst Rev 2016, 3:CD007451.

Reviewer 2 Report

General comments

Some minor revisions have to be applied.

First of all, the Authors did not introduce a Conclusions section, but some concluding considerations at the end of the Discussion section. Please add it, even if optional.

  1. Introduction

- The Authors have reported the aim of the paper, but they did not better highlight the originality and added value of their work.

- As already requested in the repvious review, it is recommendable to indicate at the end of the Introduction section the main employed characterisation techniques in order to achieve the indicated purpose.

Author Response

General comments

Some minor revisions have to be applied.

  1. First of all, the Authors did not introduce a Conclusions section, but some concluding considerations at the end of the Discussion section. Please add it, even if optional.

Response: Thanks for your suggestions. We have added the Conclusions section as required.

  1. Introduction

The Authors have reported the aim of the paper, but they did not better highlight the originality and added value of their work.

Response: Thanks for your comments. We have modified the last paragraph of Introduction to highlight the originality.

  1. As already requested in the repvious review, it is recommendable to indicate at the end of the Introduction section the main employed characterisation techniques in order to achieve the indicated purpose.

Response: Thanks for your suggestions. We have modified the Introduction section as required.
